# Chromatic Correlation Clustering, Revisited

**Qing Xiu**
School of Computer Science and Technology
Suzhou Institute for Advanced Research
University of Science and Technology of China
xiuq@mail.ustc.edu.cn

**Kai Han**[*]
School of Computer Science and Technology
Soochow University
hankai@suda.edu.cn

**Jing Tang**
The Hong Kong University of
Science and Technology (Guangzhou)
The Hong Kong University of
Science and Technology
jingtang@ust.hk

**Shuang Cui**
School of Computer Science and Technology
Suzhou Institute for Advanced Research
University of Science and Technology of China
lakers@mail.ustc.edu.cn

**He Huang**[*]
School of Computer Science and Technology
Soochow University
huangh@suda.edu.cn

## Abstract

Chromatic Correlation Clustering (CCC) (introduced by Bonchi et al. [6]) is a natural generalization of the celebrated Correlation Clustering (CC) problem. It models objects with categorical pairwise relationships by an edge-colored graph, and has many applications in data mining, social networks and bioinformatics. We show that there exists a 2.5-approximation to the CCC problem based on a Linear Programming (LP) approach, thus improving the best-known approximation ratio of 3 achieved by Klodt et al. [25]. We also present an efficient heuristic algorithm for CCC leveraging a greedy clustering strategy, and conduct extensive experiments to demonstrate the effectiveness and efficiency of our proposed algorithm.

## 1 Introduction

Clustering is the task of partitioning a set of objects into groups according to their relationships. One of the most important clustering problems is Correlation Clustering (CC), which has attracted great interests in recent years (e.g., [2, 10]). In the CC problem, the input instance is an undirected complete graph with each edge labeled as either similar or dissimilar, and the goal is to partition the graph nodes into an arbitrary number of clusters such that the number of disagreements is minimized, where a "disagreement" occurs if a dissimilar edge links two nodes in the same cluster or a similar edge links two nodes in different clusters. However, CC only models binary relationships, which is insufficient in many applications. For example, the relationships in a social network could have multiple types such as colleague, schoolmate, and family. To address this issue, Bonchi et al. [6] generalize the CC problem to the Chromatic Correlation Clustering (CCC) problem, where an edge-colored graph is used to model the multi-type relationships in practice, i.e., each edge is associated with one color and each color represents a type of relationship. The goal of CCC is to partition the graph nodes into several clusters and assign a color to each cluster to minimize the number of disagreements, where a disagreement happens in any case other than that two non-adjacent nodes belong to different clusters

---

[*]Corresponding author: Kai Han <hankai@suda.edu.cn> and He Huang <huangh@suda.edu.cn>

36th Conference on Neural Information Processing Systems (NeurIPS 2022).

or that two adjacent nodes belong to the same cluster with the cluster's color being identical to the color of the edge between these nodes. Due to its generality, CCC has wide applications including link classification, entity de-duplication, mining protein complexes in protein-protein interaction networks, and so on [3, 6, 25].

**Related work.** The original Correlation Clustering (CC) problem has been extensively studied since the seminal work of Ben-Dor et al. [5]. Bansal et al. [4] proved its NP-hardness and gave the first constant-factor approximation, which was improved to 4 in [9] by rounding the solution to a linear programming relaxation. Ailon et al. [2] proposed a simple linear-time algorithm dubbed Pivot that achieves 3-approximation, and also proposed an LP-rounding based algorithm with an improved approximation ratio of 2.5. Chawla et al. [10] further improved this ratio to 2.06 by using a more sophisticated LP-rounding technique, which nearly matches the known integrality gap of 2 [9] for the LP formulation of CC. Some studies also proposed fast heuristic algorithms for CC [16, 17, 29, 30, 34], but without providing any performance guarantee. Due to the importance of CC, a lot of its variants and related problems have been investigated in the literature [2, 7, 8, 9, 10, 13, 14, 18, 21, 22, 24, 26, 27, 31, 32, 33].

Bonchi et al. [6] initiated the studies on the Chromatic Correlation Clustering (CCC) problem and proposed a heuristic for it without a provable approximation ratio. Anava et al. [3] presented a 4-approximation algorithm for CCC based on LP-rounding, and also provided a more efficient 11-approximation algorithm dubbed Reduce and Cluster (RC) and a heuristic dubbed Deep Cluster (DC). Recently, Klodt et al. [25] showed that the classical Pivot algorithm proposed in Ailon et al. [2] for the CC problem also has an approximation ratio of 3 for CCC, and showed that the RC algorithm proposed in [3] actually has an approximation ratio of 5 for CCC. Although Pivot has the best-known approximation ratio for CCC, it works in a color-blind manner and hence has unsatisfying practical performance. Therefore, Klodt et al. [25] further proposed a heuristic dubbed Greedy Expansion (GE) which has better practical performance but without a performance guarantee. Therefore, it still remains as an open problem to find an approximation ratio for CCC tighter than the 3-approximation achieved by Pivot. CC (or CCC) also bears some similarities to the clique partitioning, multicut and k-center/k-median problems, which have been studied in some excellent proposals (e.g., [11, 15, 19, 20]). However, it is highly non-trivial (if possible) to adapt the solutions of these studies to the CCC problem.

**Our contributions.** In this paper, we propose a 2.5-approximation to the CCC problem based on an LP approach, improving the best-known ratio of 3 achieved by Klodt et al. [25] and being very close to previously known integrality gap of 2 [9]. We achieve this ratio by using a simple yet effective algorithm based on rounding the optimal solution to a linear programming relaxation of CCC, which is very different from the rounding algorithm proposed in [3]. More specifically, we first classify all vertices of the graph into several groups, with the vertices in each group colored the same, and then cluster the nodes in each group using a procedure similar to the canonical Pivot algorithm. Both the classifying phase and the clustering phase described above use the LP-solution to CCC to ensure guaranteed clustering quality. Although our LP-based clustering algorithm is randomized, it can be de-randomized to achieve a deterministc 2.5-approximation. Due to this novel clustering framework, our performance analysis is more involved than previous work to address the difficulty of counting the number of disagreements in a color-sensitive manner.

We notice that the algorithms proposed by prior studies for CCC with the best empirical performance are heuristics without performance guarantees. Therefore, we also propose a fast heuristic dubbed GreedyVote that adopts a novel greedy strategy roughly explained as follows. GreedyVote creates a cluster by firstly picking two neighboring "seed vertices", and then growing the seed set by letting the seed vertices vote for every unclustered vertex and greedily adding vertices with the largest vote counts into the seed set. We conduct extensive experiments using real-world datasets and the results strongly demonstrate the superiorities of GreedyVote in terms of both clustering quality and efficiency, compared to the state-of-the-art algorithms with or without performance guarantees.

## 2 Preliminaries

In the Chromatic Correlation Clustering (CCC) problem, an input instance consists of a finite set $L$ of colors (representing positive relationships), a special color $\gamma \notin L$ (representing negative relationship or no relationship), and an undirected complete graph $G = (V, E)$ with a color function $\phi \colon E \mapsto$

$L \cup \{\gamma\}$. The objective is to find a colored partition of the vertex set $V$ that minimizes the number of disagreements. More formally, a solution to this problem is a clustering $S = (\mathcal{C}, \Phi)$ consisting of a partition of the vertices $\mathcal{C} \colon V \mapsto \{C_1, C_2, \dots\}$ and a cluster-coloring function $\Phi \colon range(\mathcal{C}) \mapsto L$. The number of disagreements between $G$ and $S$ can be written as $\sum_{uv \in E} d_{G,S}(uv)$, where

$$d_{G,S}(uv) = \begin{cases} 0 & \phi(uv) = \gamma \wedge \mathcal{C}(u) \neq \mathcal{C}(v), \\ 0 & \phi(uv) \in L \wedge \mathcal{C}(u) = \mathcal{C}(v) \wedge \Phi(\mathcal{C}(u)) = \phi(uv), \\ 1 & \text{otherwise.} \end{cases} \tag{1}$$

For simplicity, we use $d(uv)$ as a shorthand for $d_{G,S}(uv)$. There also exists a Linear Programming (LP) relaxation of the chromatic correlation clustering problem [3], i.e., **[CCC-LP]** listed below. Let us consider an integral solution satisfying $x_{uv}^c, x_u^c \in \{0, 1\}$ for all $u \in V$, $uv \in E$, $c \in L$ to **[CCC-LP]**. The variable $x_u^c$ indicates whether vertex $u$ is in a cluster of color $c \in L$ or not (i.e., if $x_u^c = 0$ then $u$ is in the cluster colored by $c$). Similarly, the variable $x_{uv}^c$ indicates whether two vertices $u$ and $v$ are clustered into the same cluster colored by $c \in L$ (i.e., if $x_{uv}^c = 0$ then $u$ and $v$ belong to the same cluster colored by $c$). Constraint (4) guarantees that each node belongs to one colored cluster. Constraint (2) implies that the edge $uv$ can be in the cluster colored by $c \in L$ only if both vertex $u$ and $v$ are in this cluster. Constraint (3) ensures that if $uv$ and $vw$ are in the same cluster then the edge $wu$ must also be in the same cluster. According to the definition of the variable $x_{uv}^c$, we know that an edge $uv \in E$ with color $c \in L$ does not cause a disagreement only if $x_{uv}^c = 0$, and that an edge $uv$ with color $\gamma$ does not cause a disagreement only if $x_{uv}^c = 1$ for any color $c \in L$. As such, the objective function of **[CCC-LP]** consists of two parts: the first part counts the number of disagreements caused by the edges colored by $c \in L$, and the second part counts the number of disagreements caused by the edges colored by $\gamma$. Note that we can omit the variable $x_u^c$ when $|L| = 1$ and then it becomes the same LP-formulation of CC [2], and hence the integrality gap of **[CCC-LP]** is at least 2 [9].

$$\min \sum_{uv \in E \colon \phi(uv) \in L} x_{uv}^{\phi(uv)} + \sum_{uv \in E \colon \phi(uv) = \gamma} \sum_{c \in L} (1 - x_{uv}^c) \qquad \textbf{[CCC-LP]}$$

$$\text{s.t.} \quad x_{uv}^c \geq x_u^c, x_v^c \qquad\qquad\qquad \forall u, v \in V, \forall c \in L \quad (2)$$

$$x_{uv}^c + x_{vw}^c \geq x_{wu}^c \qquad\qquad\qquad \forall u, v, w \in V, \forall c \in L \quad (3)$$

$$\sum_{c \in L} x_v^c = |L| - 1 \qquad\qquad\qquad \forall v \in V \quad (4)$$

$$x_{uv}^c = x_{vu}^c \qquad\qquad\qquad \forall u, v \in V, \forall c \in L \quad (5)$$

$$x_{uv}^c, x_u^c \in [0, 1] \qquad\qquad\qquad \forall u, v \in V, \forall c \in L \quad (6)$$

## 3 An LP-based Approximation Algorithm

In this section, we present an LP-based approximation algorithm as shown in Algorithm 1, namely LP-Clustering. It is randomized and we will show how to de-randomize it in Section C. Intuitively, we aim to round the LP solution $x_v^c$ for all $v \in V, c \in L$ and $x_{uv}^c$ for all $uv \in E, c \in L$ to integers being feasible to the LP problem such that the objective value of the rounded integer solution is close to that of the optimal fractional solution as much as possible. In a nutshell, our LP-Clustering algorithm consists of two phases, including a vertex partitioning phase by rounding $x_v^c$ for all $v \in V, c \in L$ and a clustering phase for each partition by rounding $x_{uv}^c$ for all $uv \in E, c \in L$. The detailed procedures of the two phases are given as follows.

Phase 1 **Partitioning** (Lines 2–4): the vertex set $V$ is partitioned into $|L| + 1$ disjoint subsets such that $S_c = \{v \in V \colon x_v^c < \frac{1}{2}\}$ for each $c \in L$ and $O = V \setminus \left( \bigcup_{c \in L} S_c \right)$. Note that $S_c$ may be an empty set for $c \in L$. And all $S_c$ are determined.

Phase 2 **Clustering** (Lines 5–12): (i) for the partition $O$, each *outlier* $v \in O$ is assigned to a singleton cluster with an arbitrary color (Lines 5–6); (ii) and for each partition $S_c$, the algorithm iteratively chooses an unclustered vertex from $S_c$ uniformly at random as the center of a new cluster $C_k$ with color $c$, and adds each unclustered vertex $u \in S_c$ to $C_k$ with a probability of $(1 - x_{uv}^c)$ independently (Lines 7–12).

In the following, we elaborate the intuition. Recall that the variables $x_v^c$ indicates the *impossibility* of vertex $v$ being in a cluster of color $c$, while $x_{uv}^c$ represents the *distance* between vertices $u$ and $v$ with

---

**Algorithm 1:** LP-Clustering

---

**Input:** An undirected complete graph $G = (V, E)$ with a color function $\phi \colon E \mapsto L \cup \{\gamma\}$, the LP solution $x$, i.e., $\{x_v^c \colon \forall v \in V, \forall c \in L\}$ and $\{x_{uv}^c \colon \forall uv \in E, \forall c \in L\}$

**Output:** A clustering $\{C_1, C_2, \dots\}$ and a cluster-coloring function $\Phi$

1   Initialize $k \leftarrow 0$ and $O \leftarrow V$;

2   **foreach** $c \in L$ **do**

3      Let $S_c$ be the subset of nodes in $O$ with value $x_v^c < \frac{1}{2}$, i.e., $S_c \leftarrow \{v \in O \colon x_v^c < \frac{1}{2}\}$;

4      Remove $S_c$ from $O$;

5   **foreach** $v \in O$ **do**

6      Create a new cluster $C_k = \{v\}$ by increasing $k$ by 1, and set $\Phi(C_k)$ as an arbitrary color;

7   **foreach** $c \in L$ **do**

8      **while** $S_c \neq \emptyset$ **do**

9         Pick a vertex $v \in S_c$ uniformly at random;

10        Create a new cluster $C_k \leftarrow \{v\}$ by increasing $k$ by 1, and set $\Phi(C_k) \leftarrow c$;

11       **foreach** $u \in S_c \setminus \{v\}$ **do** add $u$ to $C_k$ with probability $1 - x_{uv}^c$;

12       Remove $C_k$ from $S_c$;

13   **return** $\{C_1, C_2, \dots, C_k\}, \Phi$

---

respect to color $c$. Based on this observation, our LP-Clustering algorithm first determines the *vertex color* for each vertex $v$ by rounding $x_v^c$, i.e., the color of the cluster to which vertex $v$ belongs with the maximum likelihood. That is, $v$ will be assigned to a cluster with color $c$ if $x_v^c < 1/2$, which ensures that $x_v^c \leq x_v^{\bar{c}}$ for any $\bar{c} \in L$ implied by Observation 4.1. This is equivalent to round $x_v^c$ to 0 and $x_v^{\bar{c}}$ to 1 for every $\bar{c} \neq c$. Meanwhile, we call a vertex $v$ *outlier* if $x_v^c \geq 1/2$ for every $c \in L$, which is far away from any other vertex $u$ as to every color $c$ due to the fact that $x_{uv}^c \geq x_v^c \geq 1/2$. Then, for each outlier $v$, we just arbitrarily choose a color $c$ (e.g., the one with the smallest value of $x_v^{\bar{c}}$) to round $x_v^c$ to 0 and $x_v^{\bar{c}}$ to 1 for every $\bar{c} \neq c$. It is trivial to see that the rounding of $x_v^c$ actually utilizes a greedy strategy such that for each $v$, the smallest $x_v^c$ is rounded to 0 while the others are rounded to 1.

Next, we finalize the clustering by rounding $x_{uv}^c$ with respect to the rounded value of $x_v^c$ for each $v \in V$ and $c \in L$. Specifically, for each outlier $v \in O$, assigning it to a singleton cluster, i.e., a cluster containing this node only, is equivalent to round $x_{uv}^c$ to 1 for every $u \in V \setminus \{v\}$ and $c \in L$. Since $x_{uv}^c \geq x_v^c \geq 1/2$, such a rounding ensures the corresponding cost of the integer solution within twice of that of the fractional solution. Moreover, for the subset $S_c$ of $V$ containing vertices with color $c$, we iteratively select an unclustered vertex $v$ from $S_c$ uniformly at random and create a new cluster $C_k = \{v\}$, until every vertex in $S_c$ is included in some cluster. Then, we add each unclustered vertex $u \in S_c$ to $C_k$ with a probability of $(1 - x_{uv}^c)$, i.e., rounding $x_{uv}^c$ to 0 with a probability of $(1 - x_{uv}^c)$ and to 1 with the other probability so that the rounded $x_{uv}^c$ in expectation is exactly $x_{uv}^c$. Such a rounding again ensures the corresponding cost of the integer solution within a constant factor of that of the fractional solution.

In the following theorem, we give our main result that the LP-Clustering algorithm achieves a 2.5-approximation in expectation for the Chromatic Correlation Clustering (CCC) problem, which improves the best-known factor of 3 achieved by Klodt et al. [25].

**Theorem 3.1.** *The LP-Clustering algorithm achieves an approximation of* 2.5 *in expectation for the CCC problem.*

Next in Section 4, we prove our main theoretical result in Theorem 3.1, where the full proof of a core lemma (i.e., Lemma 4.3) requires a non-trivial analysis and can be found in Appendix A.

## 4   Performance Analysis

### 4.1   Overview

For our randomized rounding approach based on the LP solution, the expected total number of disagreements is $\mathbb{E}[\sum_{uv \in E} d(uv)]$, where $d(uv)$ is defined in (1). Similarly, the objective function in

**[CCC-LP]** can be written as $\sum_{uv \in E} lp(uv)$, where $lp(uv)$ denotes the cost on edge $uv$, i.e.,

$$lp(uv) := \begin{cases} x_{uv}^{\phi(uv)}, & \text{if } \phi(uv) \in L, \\ \sum_{c \in L}(1 - x_{uv}^c) & \text{if } \phi(uv) = \gamma. \end{cases} \qquad (7)$$

Therefore, to prove Theorem 3.1, it suffices to show that

$$\mathbb{E}\Big[ \sum_{uv \in E} d(uv) \Big] \leq \frac{5}{2} \cdot \sum_{uv \in E} lp(uv). \qquad (8)$$

Since the cost is characterized by each edge $uv$, according to the partitioning of $V = O \cup \big(\bigcup_{c \in L} S_c\big)$, we partition the edge set $E$ into $|L| + 2$ disjoint subsets $E_o$, $E_i$ and $E_c$ for $c \in L$, i.e.,

$$\begin{aligned} E_o &:= \{uv \in E \colon v \in O\}, \\ E_i &:= \{uv \in E \colon u \in S_c, v \in S_{\bar{c}}, c \neq \bar{c}\}, \\ E_c &:= \{uv \in E \colon u \in S_c, v \in S_c\}, \ \forall c \in L. \end{aligned}$$

Clearly, $E = E_o \cup E_i \cup \big(\bigcup_{c \in L} E_c\big)$. In particular, $E_o$ is the set of edges incident on outliers in $O$; $E_i$ is the set of edges between non-outliers cross two vertex partitions $S_c$ and $S_{\bar{c}}$; and $E_c$ is the set of edges between vertices within the same vertex partition $S_c$. In what follows, we establish the relation between LP-Clustering and LP (i.e., the objective function of **[CCC-LP]**) in terms of the cost on different category of edge.

For each edge $uv \in E_o$, it is an inter-cluster edge. By (1), we can get that $d(uv) = 1$ if $\phi(uv) \in L$ and $d(uv) = 0$ otherwise. Meanwhile, constraint (2) ensures that $x_{uv}^c \geq x_v^c \geq \frac{1}{2}$ for every $u \neq v$ and $c \in L$ when $v$ is an outlier. This implies that $lp(uv) = x_{uv}^{\phi(uv)} \geq \frac{1}{2}$ if $\phi(uv) \in L$ and $lp(uv) \geq 0$ otherwise. We immediately have $d(uv) \leq 2 \cdot lp(uv)$.

For each edge $uv \in E_i$, it is again an inter-cluster edge, indicating that $d(uv) = 1$ if $\phi(uv) \in L$ and $d(uv) = 0$ otherwise. To derive $lp(uv)$, we first present a useful property of the fractional solution.

**Observation 4.1.** *Given a vertex $v \in V$ and any two distinct colors $c_1, c_2 \in L$ then $x_v^{c_1} + x_v^{c_2} \geq 1$.*

In fact, Observation 4.1 can be easily obtained by contradiction, since otherwise $\sum_{c \in L} x_v^c < |L| - 1$, which contradicts constraint (4). Denote by $c_u$ the vertex color of $u$ such that $x_u^{c_u} < \frac{1}{2}$. Then, by Observation 4.1, we have $x_u^c \geq \frac{1}{2}$ for any $c \neq c_u$. As a consequence, $lp(uv) = x_{uv}^{\phi(uv)} \geq x_u^{\phi(uv)} \geq \frac{1}{2}$ if $\phi(uv) \in L \setminus \{c_u\}$, $lp(uv) = x_{uv}^{\phi(uv)} \geq x_v^{c(u)} \geq \frac{1}{2}$ if $\phi(uv) = c_u \neq c_v$, and $lp(uv) \geq 0$ otherwise. This concludes that $d(uv) \leq 2 \cdot lp(uv)$.

Finally, if we show that $\mathbb{E}[\sum_{uv \in E_c} d(uv)] \leq \alpha \cdot \sum_{uv \in E_c} lp(uv)$ for every $c \in L$ and some $\alpha \geq 2$, combing with the above analysis, we immediately derive an upper bound on the expected cost of our output clustering, i.e.,

$$\begin{aligned} \mathbb{E}\Big[ \sum_{uv \in E} d(uv) \Big] &= \sum_{uv \in E_o} d(uv) + \sum_{uv \in E_i} d(uv) + \sum_{c \in L} \mathbb{E}\Big[ \sum_{uv \in E_c} d(uv) \Big] \\ &\leq 2 \cdot \sum_{uv \in E_o} lp(uv) + 2 \cdot \sum_{uv \in E_i} lp(uv) + \alpha \cdot \sum_{c \in L} \sum_{uv \in E_c} lp(uv) \\ &\leq \alpha \cdot \sum_{uv \in E} lp(uv). \end{aligned}$$

We thus obtain the following lemma.

**Lemma 4.1.** *Given $\alpha \geq 2$, if the clustering returned by LP-Clustering satisfies for every $c \in L$,*

$$\mathbb{E}\Big[ \sum_{uv \in E_c} d(uv) \Big] \leq \alpha \cdot \sum_{uv \in E_c} lp(uv),$$

*then LP-Clustering achieves an approximation guarantee of $\alpha$ for chromatic correlation clustering.*

Therefore, to get the approximation ratio of LP-Clustering, by Lemma 4.1, we just need to show the relation between $\mathbb{E}[\sum_{uv \in E_c} d(uv)]$ and $\sum_{uv \in E_c} lp(uv)$ for each color $c \in L$.

## 4.2 More Detailed Analysis

In what follows, we focus on the elementary analysis over $E_c$ for every $c \in L$. For each given $S_c$, LP-Clustering iteratively chooses some vertices to create a new cluster. Denote by $S_c^t$ the unclustered vertex set at the beginning of iteration $t$, by $E_c^t$ the induced edge set by $S_c^t$ containing all the edges with both endpoints in $S_c^t$, and by $C_c^t$ the cluster created in this iteration. With removing $C_c^t$ from $S_c^t$, the edges incident on vertices in $C_c^t$, i.e., $E_c^t \setminus E_c^{t+1} = \{uv \in E_c \colon u, v \in S_c^t, (u \in C_c^t \vee v \in C_c^t)\}$, are also removed from $E_c$. Then, the number of disagreements of these edges (i.e., $E_c^t \setminus E_c^{t+1}$), denoted by $D_t$, can be derived by

$$D_t = \sum_{uv \in E_c^t \colon \phi(uv)=c} \mathbb{1}(\mathcal{E}_{u-v}^t) + \sum_{uv \in E_c^t \colon \phi(uv)\in L\setminus\{c\}} \left(\mathbb{1}(\mathcal{E}_{u-v}^t) + \mathbb{1}(\mathcal{E}_{u+v}^t)\right) + \sum_{uv \in E_c^t \colon \phi(uv)=\gamma} \mathbb{1}(\mathcal{E}_{u+v}^t),$$

where $\mathbb{1}(\mathcal{E})$ denotes the indicator function of the event $\mathcal{E}$, $\mathcal{E}_{u-v}^t$ denotes the event that only one of $u$ and $v$ is in $C_c^t$, and $\mathcal{E}_{u+v}^t$ denotes the event that both $u$ and $v$ are in $C_c^t$. Similarly, the LP cost on $E_c^t$, denoted by $L_t$, can be derived by

$$L_t = \sum_{uv \in E_c^t \colon \phi(uv)\in L} \left(\mathbb{1}(\mathcal{E}_{u-v}^t)+\mathbb{1}(\mathcal{E}_{u+v}^t)\right)x_{uv}^{\phi(uv)} + \sum_{uv \in E_c^t \colon \phi(uv)=\gamma} \left(\mathbb{1}(\mathcal{E}_{u-v}^t)+\mathbb{1}(\mathcal{E}_{u+v}^t)\right)\sum_{i\in L}(1-x_{uv}^i).$$

Let $T$ be the stopping time such that all vertices in $S_c$ are clustered after $T$ iterations. Then, we know that $\sum_{t=0}^{T} D_t = \sum_{uv \in E_c} d(uv)$ and $\sum_{t=0}^{T} L_t = \sum_{uv \in E_c} lp(uv)$, since each edge in $E_c$ can only be removed exactly once. If we show that $\mathbb{E}[D_t] \leq \alpha\mathbb{E}[L_t]$ for all $t$, then we know that $Z_s = \sum_{t=0}^{s}(\alpha L_t - D_t)$ is a submartingale (i.e., $\mathbb{E}[Z_{s+1} \mid Z_s] \geq Z_s$) and hence $\mathbb{E}[Z_T] \geq \mathbb{E}[Z_0]$, which yields that

$$\alpha \sum_{uv \in E_c} lp(uv) - \mathbb{E}\Big[\sum_{uv \in E_c} d(uv)\Big] = \mathbb{E}[Z_T] \geq \mathbb{E}[Z_0] = \alpha\mathbb{E}[L_0] - \mathbb{E}[D_0] \geq 0.$$

To get $\mathbb{E}[D_t] \leq \alpha\mathbb{E}[L_t]$, it suffices to show its conditional version that $\mathbb{E}[D_t \mid S_c^t] \leq \alpha\mathbb{E}[L_t \mid S_c^t]$ for all $S_c^t \subset S_c$. Given $S_c^t$, each vertex $w \in S_c^t$ is picked as the center of the cluster $C_c^t$ with a probability of $\frac{1}{|S_c^t|}$. Thus, given an event $\mathcal{E}$, we have

$$\mathbb{E}[\mathbb{1}(\mathcal{E}) \mid S_c^t] = \Pr[\mathcal{E} \mid S_c^t] = \frac{1}{|S_c^t|} \sum_{w\in S_c^t} \Pr[\mathcal{E} \mid center = w, S_c^t].$$

For simplicity, let

$$d(uv \mid w) := \begin{cases} \Pr[\mathcal{E}_{u-v}^t \mid center = w, S_c^t], & \text{if } \phi(uv) = c, \\ \Pr[\mathcal{E}_{u-v}^t \mid center = w, S_c^t] + \Pr[\mathcal{E}_{u+v}^t \mid center = w, S_c^t], & \text{if } \phi(uv) \in L \setminus \{c\}, \\ \Pr[\mathcal{E}_{u+v}^t \mid center = w, S_c^t], & \text{if } \phi(uv) = \gamma. \end{cases}$$

Moreover, according to the definitions of $\mathcal{E}_{u-v}^t$ and $\mathcal{E}_{u+v}^t$, we have

$$\Pr[\mathcal{E}_{u-v}^t \mid center = w, S_c^t] = (1 - x_{vw}^c)x_{uw}^c + (1 - x_{uw}^c)x_{vw}^c,$$
$$\Pr[\mathcal{E}_{u+v}^t \mid center = w, S_c^t] = (1 - x_{uw}^c)(1 - x_{vw}^c).$$

Note that by setting $x_{uu}^c = 0$ for each $u \in S_c^t$, the above equations also hold when $w = u$ or $w = v$. As a consequence, we have

$$d(uv \mid w) = \begin{cases} (1 - x_{vw}^c)x_{uw}^c + (1 - x_{uw}^c)x_{vw}^c, & \text{if } \phi(uv) = c, \\ 1 - x_{uw}^c x_{vw}^c, & \text{if } \phi(uv) \in L \setminus \{c\}, \\ (1 - x_{uw}^c)(1 - x_{vw}^c), & \text{if } \phi(uv) = \gamma. \end{cases}$$

Putting it together, we can express $\mathbb{E}[D_t \mid S_c^t]$ as

$$\mathbb{E}[D_t \mid S_c^t] = \frac{1}{|S_c^t|} \sum_{w\in S_c^t} \sum_{uv\in E_c^t} d(uv \mid w) = \frac{1}{2|S_c^t|} \sum_{u,v,w\in S_c^t} d(uv \mid w),$$

where $d(uu \mid w) := 0$ for all $u, w \in S_c^t$. Similarly, let

$$lp(uv \mid w) := \begin{cases} (1 - x_{uw}^c x_{vw}^c)x_{uv}^{\phi(uv)}, & \text{if } \phi(uv) \in L, \\ (1 - x_{uw}^c x_{vw}^c)\sum_{i\in L}(1 - x_{uv}^i), & \text{if } \phi(uv) = \gamma. \end{cases}$$

---

**Algorithm 2:** GreedyVote

---

**Input:** An undirected graph $G = (V, E)$ with a color function $\phi : E \to L$, parameters $\epsilon$ and $m$.

**Output:** A clustering $\{S_1, S_2, \ldots, S_k\}$ and a cluster-coloring function $\Phi$.

**1** Initialize $k = 0$;

**2 while** $E \neq \emptyset$ **do**

**3**     Let $M$ be a set of $m$ edges sampled with replacement from $E$;

**4**     Let $uv$ be the edge in $M$ that maximizes $\frac{|N_{\phi(uv)}(u) \cap N_{\phi(uv)}(v)|}{|N(u) \cup N(v)|}$;

**5**     Create a new cluster $S_k = \{u, v\}$ by increasing $k$ by one, and remove $S_k$ from $V$;

**6**     **while** $V \neq \emptyset$ **do**

**7**        Let each vertex in $S_k$ vote for every vertex in $V$ such that vote 0.5 when there is an edge between the two vertices, and an extra 0.5 if the edge color is $\phi(uv)$;

**8**        Let $w \in V$ be the vertex with the largest vote count and let $\ell$ be the vote count of $w$;

**9**        **if** $\ell \geq \epsilon|S_k|$ **then** Add $w$ to $S_k$ and remove $w$ from $V$;

**10**        **else Break**;

**11**     Let $c$ be the most frequent color of the edges with both endpoints in $S_k$, and set $\Phi(S_k) = c$;

**12**     Remove all edges in $\{uv \in E | u \in S_k\}$ from $E$;

**13 foreach** $v \in V$ **do**

**14**     Create a new cluster $S_k = \{v\}$ by increasing $k$ by one;

**15**     Let $\Phi(S_k) = c$ where $c$ is an arbitrary color in $L$;

**16 return** $\{S_1, S_2, ..., S_k\}, \Phi$

---

Then, $\mathbb{E}[L_t \mid S_c^t]$ can be expressed as

$$\mathbb{E}[L_t \mid S_c^t] = \frac{1}{|S_c^t|} \sum_{w \in S_c^t} \sum_{uv \in E_c^t} lp(uv \mid w) = \frac{1}{2|S_c^t|} \sum_{u,v,w \in S_c^t} lp(uv \mid w),$$

where $lp(uu \mid w) := 0$ for all $u, w \in S_c^t$. Observing that $d(uv \mid w) \leq \alpha \cdot lp(uv \mid w)$ does not always hold, thanks to the symmetry, we consider $D(uvw) := d(uv \mid w) + d(vw \mid u) + d(wu \mid v)$ as a whole. That is,

$$\mathbb{E}[D_t \mid S_c^t] = \frac{1}{6|S_c^t|} \sum_{u,v,w \in S_c^t} \big(d(uv \mid w) + d(vw \mid u) + d(wu \mid v)\big) = \frac{1}{6|S_c^t|} \sum_{u,v,w \in S_c^t} D(uvw).$$

Similarly, letting $L(uvw) := lp(uv \mid w) + lp(vw \mid u) + lp(wu \mid v)$, we have

$$\mathbb{E}[L_t \mid S_c^t] = \frac{1}{6|S_c^t|} \sum_{u,v,w \in S_c^t} \big(lp(uv \mid w) + lp(vw \mid u) + lp(wu \mid v)\big) = \frac{1}{6|S_c^t|} \sum_{u,v,w \in S_c^t} L(uvw).$$

Therefore, it suffices to show that $D(uvw) \leq \alpha L(uvw)$. We thus obtain the following lemma.

**Lemma 4.2.** *Given a vertex set $S_c$, if it always holds that $D(uvw) \leq \alpha L(uvw)$ for all $u, v, w \in S_c^t$, we have $\mathbb{E}[\sum_{uv \in E_c} d(uv)] \leq \alpha \cdot \sum_{uv \in E_c} lp(uv)$*

Finally, the following lemma gives such a relation.

**Lemma 4.3.** *For all $c \in L$ and $u, v, w \in S_c^t$, we have $D(uvw) \leq 2.5L(uvw)$.*

The proof of Lemma 4.3 is based on a nontrivial case analysis with respect to the the edge color of $\phi(uv)$, $\phi(vw)$ and $\phi(wu)$, which can be found in Appendix A. Now, putting it together of Lemmas 4.1–4.3 concludes Theorem 3.1.

## 5   A Greedy Clustering Heuristic

For the CCC problem, prior studies have shown that some heuristics without any performance guarantees (e.g., the Deep Cluster algorithm in [3] and the Greedy Expansion algorithm in [25]) can empirically perform much better than the existing algorithms with provable approximation ratios including the Pivot algorithm with the best-known ratio of 3. This motivates us to design a fast heuristic with better performance in practice, namely the GreedyVote algorithm shown in Algorithm 2. Roughly speaking, GreedyVote repeats in iterations and creates a cluster $S_k$ in the $k$th iteration according to the following two phases of operations:

Phase 1 **Seed selection** (Lines 3–5): Randomly select $m$ candidate edges and pick one of them (denoted by $uv$) to create a "seed set" $S_k = \{u, v\}$;

Phase 2 **Vote to grow** (Lines 6–10): Each vertex in the seed set $S_k$ votes for (or scores) every currently unclustered vertex, and the vertex $w$ with the largest vote count is added into $S_k$ if satisfying the condition in Line 9. Repeat this process until there are no qualified vertices.

In the sequel, we provide more detailed explanations on the two phases mentioned above. The purpose of the "seed selection" phase is to identify two neighboring nodes that are "most likely" to be in the same cluster of a good solution. To achieve this, we use a metric inspired by the well-known Jaccard Similarity to measure the goodness of $u$ and $v$ being clustered together, i.e., $\frac{|N_{\phi(uv)}(u) \cap N_{\phi(uv)}(v)|}{|N(u) \cup N(v)|}$ in Line 4, where $N(u)$ denotes the set of all neighbors of $u$ and $N_c(u) : \forall c \in L$ denotes the set of vertices adjacent to $u$ by $c$-colored edges. Instead of checking every pair of neighboring nodes for this metric, we randomly check $m$ edges and pick the best one among them to save running time.

Given the initial seed set constructed by Phase 1, the purpose of the "vote to grow" phase is to identify some qualified vertices and add them into $S_k$ one by one. The rational for judging whether a candidate vertex $w$ should be added into $S_k$ or not is roughly explained as follows. Let $|E_c(S_k, w)|$ denote the number of edges with color $c$ ($\forall c \in L$) linking $w$ and any vertex in $S_k$, and let $error(S_k)$ denote the number of disagreements caused by edges with both endpoints in $S_k$. Let $u$ and $v$ be the initial two seed vertices in $S_k$ found by Line 4, and suppose that $S_k$ is assigned the color of $\phi(uv)$. Then, not adding $w$ into $S_k$ would cause at least $Er_1 = error(S_k) + \sum_{c \in L} |E_c(S_k, w)|$ disagreements, while adding $w$ into $S_k$ would bring $Er_2 = error(S_k) + |S_k| - |E_{\phi(uv)}(S_k, w)|$ disagreements. Clearly, adding $w$ into $S_k$ would be beneficial only when $Er_2 - Er_1 = |S_k| - |E_{\phi(uv)}(S_k, w)| - \sum_{c \in L} |E_c(S_k, w)| \leq 0$, or equivalently $|E_{\phi(uv)}(S_k, w)| + \sum_{c \in L} |E_c(S_k, w)| \geq |S_k|$. Actually, it is not hard to verify that the voting mechanism in Line 7 results in the vote count of any candidate vertex $w$ being equal to $0.5|E_{\phi(uv)}(S_k, w)| + 0.5 \sum_{c \in L} |E_c(S_k, w)|$, while we replace $|S_k|$ by $\epsilon|S_k|$ in the if-condition of Line 9 to offset the inaccuracy of counting the number of disagreements by considering every vertex individually.

When Phase 1 and Phase 2 described above finishes, no qualified vertices can be added into the current clusters. So GreedyVote creates a cluster for each leftover vertex and assign it an arbitrary color in $L$ (Lines 13-15). It can be seen that GreedyVote can be implemented to run in $\mathcal{O}(|E| + \Delta|V|)$ time for any constant $m$, where $\Delta$ is the maximum degree of $G$ (see Appendix B for a formal proof).

## 6  Performance Evaluation

In this section, we compare our GreedyVote (GV) algorithm (i.e., Algorithm 2) with six state-of-the-art algorithms for the CCC problem, including: (1) The Pivot Algorithm [2]; (2) The Reduce and Cluster (RC) Algorithm [3]; (3) The Deep Cluster (DC) algorithm [3]; (4) The Chromatic Ball (CB) algorithm [6]; (5) The Greedy Expansion (GE) Algorithm [25]; and (6) The modified greedy expansion (GER) Algorithm [25]. Among these algorithms, Pivot and RC have approximation ratios of 3 and 5, respectively, and all other algorithms are heuristics. In our experiments, we set $\epsilon = 0.55$ and $m = 10$ for the GV algorithm, and follow all the parameter settings (if any) of other algorithms according to their proposals. We implement the evaluated algorithms using Python and also use the public code of Klodt et al. [25] for implementation. All our experiments are conducted on an Intel(R) Xeon(R) Gold 6126 CPU @ 2.60GHz with 128GB of RAM. Each algorithm is run for 50 times and the average result is reported.

We use ten real-world datasets from different domains in our experiments, as listed in Table 1. Among these datasets, Facebook1, Facebook2, Lastfm and Twitter are social network graphs downloaded from [28], where vertices represent users and edges represent friendships. Each user has several properties representing the "social circles" interested by the user. Following the setting in Anava et al. [3], we assign each social circle a color and set the color of each edge $uv$ as the color of $u$ and $v$'s common social circle (if there are multiple common circles then choose an arbitrary one). The DAWN dataset [1][25] models a drug abuse warning network where vertices represent drugs and edges indicate that they were used together before an emergency room visit. Following [25], the color of the edge represents the most common outcome of the visit. The Cooking dataset [23][25] models an ingredient co-occur network where vertices represent ingredients and edges indicate that they co-occur in some recipes. Following [25], the color of the edge represents the most common

Table 1: Characteristics of the datasets used in the experiments.

| Datasets | $|V|$ | $|E|$ | $|L|$ |
|---|---|---|---|
| Facebook1 | 2871 | 62334 | 193 |
| Facebook2 | 3978 | 78011 | 46 |
| Lastfm | 7624 | 27806 | 10 |
| Twitter | 74402 | 1003951 | 99 |
| DAWN | 2109 | 96047 | 10 |
| Cooking | 6714 | 479921 | 21 |
| String1 | 17458 | 419190 | 4 |
| String2 | 18152 | 401582 | 4 |
| DBLP | 73624 | 835414 | 100 |
| MAG | 80198 | 237261 | 11 |

Table 2: Compare all algorithms on the Number of Disagreements (NOD), where the reported data of all algorithms except GV are normalized by those of GV. Lowest values are highlighted in bold.

| Datasets | Pivot [2] | RC [3] | DC [3] | CB [6] | GE [25] | GER [25] | GV (ours) |
|---|---|---|---|---|---|---|---|
| Facebook1 | 1.486 | 1.283 | 1.218 | 1.246 | 1.015 | 1.022 | **38,773** |
| Facebook2 | 1.419 | 1.305 | 1.236 | 1.220 | 1.015 | 1.025 | **52,164** |
| Lastfm | 1.442 | 1.267 | 1.267 | 1.056 | 1.013 | 1.034 | **23,046** |
| Twitter | 1.449 | 1.138 | 1.134 | 1.050 | 1.004 | 1.007 | **925,028** |
| DAWN | 1.460 | 1.236 | 1.253 | 1.115 | **0.995** | 1.035 | 84,492 |
| Cooking | 1.306 | 1.110 | 1.092 | 1.039 | 1.004 | 1.001 | **456435** |
| String1 | 1.535 | 1.219 | 1.229 | 1.072 | 1.018 | 1.038 | **359,202** |
| String2 | 1.424 | 1.409 | 1.214 | 1.429 | 1.018 | 1.017 | **112,340** |
| DBLP | 1.563 | 1.143 | 1.103 | 1.141 | 1.014 | 1.040 | **540,683** |
| MAG | 1.568 | 1.294 | 1.286 | 1.078 | 1.021 | 1.040 | **127,383** |

cuisine. String1 and String2 [12] are networks containing protein-protein interactions, where vertices represent proteins and edges represent pairs of interacting proteins. Following [6][25], we assign each protein-interaction a color and choose an arbitrary color for an edge if there are multiple interactions between two proteins. Finally, DBLP and MAG [25] are co-authorship networks where vertices represent authors and each edge is labeled (colored) by the most frequent journal or conference that two authors co-authored.

In Table 2, we list the Number of Disagreements (NOD) of all the evaluated algorithms, where the reported data of all algorithms except GV are normalized by those of GV. It can be seen that Pivot performs the worst, as it works in a color-blind way. Although RC has a worse approximation ratio than Pivot, it performs better than Pivot but worse than CB and DC, while CB and DC have similar performance on NOD. The GER algorithm outperforms the Pivot, RC, CB and DC algorithms on NOD, while GE even performs better than GER. However, it can be seen that our GV algorithm outperforms the best baseline GE in the literature on all the datasets except DAWN, while the gap between GV and GE is no more than 0.5% on the DAWN dataset. This demonstrates the superiority of our GV algorithm on the metric of NOD.

In Table 3, we compare the running time of the implemented algorithms. It can be seen that Pivot and GE generally incur the shortest and longest running time, respectively. Compared to GE, GV runs faster on all datasets except String2, MAG and Cooking, with the performance gain ranging from 11.0% to 78.8%, while the average performance gain is 27%. This can be explained by the fact that GV has a better time complexity of $\mathcal{O}(|E| + \Delta|V|)$ than the $\mathcal{O}(\Delta_L|E| + \Delta|V|)$ time complexity of GE, where $\Delta_L$ is the maximum number of distinct colors of the edges incident to any vertex.

In summary, the experimental results demonstrate that, our GreedyVote algorithm outperforms all the implemented state-of-the-art algorithms (with or without performance guarantee) in terms of the objective function value of chromatic correlation clustering, and it also runs significantly faster than GE in general, where GE is the heuristic proposed by Klodt et al. [25] with the best-known empirical performance on clustering quality.

Table 3: Compare all algorithms on the running time (seconds).

| Datasets | Pivot [2] | RC [3] | DC [3] | CB [6] | GE [25] | GER [25] | GV (ours) |
|----------|-----------|--------|--------|--------|---------|----------|-----------|
| Facebook1 | 0.06 | 0.43 | 0.64 | 0.08 | 0.86 | 0.80 | 0.54 |
| Facebook2 | 0.07 | 0.55 | 0.89 | 0.10 | 0.93 | 1.04 | 0.66 |
| Lastfm | 0.03 | 0.25 | 0.34 | 0.06 | 0.54 | 0.52 | 0.41 |
| Twitter | 0.87 | 6.71 | 8.78 | 2.16 | 17.27 | 10.75 | 15.00 |
| DAWN | 0.06 | 0.57 | 1.16 | 0.12 | 1.31 | 1.01 | 1.18 |
| Cooking | 0.24 | 3.28 | 4.92 | 1.17 | 8.73 | 4.27 | 9.54 |
| String1 | 0.23 | 2.77 | 4.72 | 0.78 | 8.99 | 5.90 | 5.85 |
| String2 | 0.57 | 3.10 | 3.35 | 0.60 | 2.97 | 5.36 | 3.37 |
| DBLP | 0.97 | 6.78 | 7.03 | 2.15 | 18.81 | 9.74 | 10.52 |
| MAG | 0.48 | 2.54 | 2.90 | 0.66 | 3.96 | 4.53 | 3.99 |

## 7 Conclusion

We have revisited the Chromatic Correlation Clustering (CCC) problem, which generalizes the correlation clustering problem and has many important applications in machine learning and data mining. For this problem, prior studies have proposed both approximation algorithms with provable performance ratios and heuristics with better practical performance. We have proposed a 2.5-approximation to CCC based on rounding the solution to a linear programming relaxation of CCC, and also a heuristic algorithm based on a "greedy voting" strategy. Our theoretical analysis and extensive experiments on real-world datasets demonstrate that our approach outperforms the existing ones both on the theoretical approximation ratio and on the practical performance in terms of time efficiency and clustering quality.

## Acknowledgement

Kai Han's work is partially supported by National Natural Science Foundation of China (NSFC) under Grant No. 62172384. He Huang's work is partially supported by National Natural Science Foundation of China (NSFC) under Grant No. U20A20182 and Grant No. 61873177. The work of Qing Xiu and Shuang Cui is done under the guidance of their supervisor: Kai Han.

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
