# OpenReview forum: "Chromatic Correlation Clustering, Revisited"
_NeurIPS.cc/2022/Conference — NeurIPS 2022 Accept_

### Official Review · Reviewer_FWeo · 2022-06-22

**Rating:** 8
**Confidence:** 5
**Soundness:** 4 excellent
**Presentation:** 4 excellent
**Contribution:** 4 excellent

**Summary:**

The authors propose improved algorithms for the chromatic correlation clustering (CCC), which is a generalization of the correlation clustering problem to multiple labels. The authors present a 2.5 approximation algorithm for this problem based on rounding of a linear program; this bound is tighter than the best-known approximation of 3. The authors also propose a heuristic motivated by their rounding procedure. Experimental results show that the proposed heuristic outperforms existing heuristics and approximation algorithms in minimizing the objective function.

**Questions:**

Have you implemented LP-Clustering?
Can you provide experimental results for LP-Clustering?
How does the improvement in performance in the objective function translate to a practical benefit? GE and GER have very similar performance.

**Limitations:**

Yes

**Strengths And Weaknesses:**

Strengths:
- Rigorous theoretical results and an improved approximation bound to a problem that has been of interest in the past 10 years
- Proposed heuristic outperforms existing alternatives
- Compelling mix of theoretical and empirical results
- Supplementary material makes it very easy to reproduce the results in the paper

Weaknesses:
- Lack of experimental results for the approximation algorithm
- Impact is somewhat unclear. How does the improvement in performance in the objective function translate to a practical benefit?

---

> ### Author Response · Authors · 2022-08-02
> **Response to reviewer FWeo (part 1/1)**
>
> # Reviewer FWeo:
>
> Thank you very much for your great comments. Please find our detailed responses in the following:
>
> > Have you implemented LP-Clustering? Can you provide experimental results for LP-Clustering? How does the improvement in performance in the objective function translate to a practical benefit? GE and GER have very similar performance.
>
> **Response**: We greatly appreciate the reviewer for the comments. It is a nice suggestion. Unfortunately, we find that solving LP for CCC requires prohibitive amount of memory and running time in practice, which are polynomial in theory though. As a result, we are unable to obtain the results for the LP-based approaches due to equipment limitations. To our knowledge, no existing work provides empirical results for the LP-based approaches including [2, 3, 10, 13, 17, 18], which indicate that such approaches are more of theoretical interests.

---

### Official Review · Reviewer_4VF8 · 2022-07-05

**Rating:** 5
**Confidence:** 4
**Soundness:** 2 fair
**Presentation:** 3 good
**Contribution:** 2 fair

**Summary:**

In this paper, the authors propose two algorithms to tackle the chromatic correlation clustering problem. The first algorithm is a randomized algorithm that outputs a feasible solution to the chromatic correlation clustering problem that is a 2.5-approximation on average. The second procedure is instead a heuristic. The authors provide a theoretical analysis regarding the first method as well as numerical experiments that show the effectiveness of the second algorithm.

**Questions:**

- Is the analysis in section 4 tight?
- Why are we guaranteed that Algorithm 2 does not get stuck in the first while loop? In fact, as long as E has at least one edge we stay in the loop, however we need at least m edges to perform all the operations inside of the while loop (and then remove edges from E). It looks to me like the algorithm could arrive at a point where E is not empty but contains less than m edges. This would result in getting stuck in the loop.

**Limitations:**

The discussion is adequate for this theoretical work.

**Strengths And Weaknesses:**

The paper is overall well-written and easy to read. The authors argue well why this problem is relevant. On the one hand, they improve by 0.5 the previously best ratio for a chromatic correlation clustering approximation algorithm. On the other one, their heuristic algorithm outperforms the current state-of-the-art heuristics on the tested data sets.

There are four main weaknesses:
1. The approximation algorithm (which should be the main result of this paper) has not been implemented and tested on the data sets. It is somehow disappointing to only see computational experiments for the heuristic approach. In particular, it would have been nice to see how tight the approximation ratio is in practice.
2. The statement in the abstract about the approximation algorithm is misleading. The information that the approximation algorithm is a randomized algorithm appears for the first time in Section 3. Moreover, the authors show that the approximation ration is attained in expectation and therefore not always. This is in contradiction to the (deterministic) statements in the abstract and introduction.
3. The choice of references a bit odd. It would be nice if the vast literature on the clique partitioning/multicut problem was cited. In this literature, the correlation clustering ILP is studied in great detail. See e.g. Groetschel & Wakabayashi, Chopra & Rao, Deza & Groetschel & Laurent, Bandelt et. al. ... (and references therein).
4. The approximation algorithm (Algorithm 1) is only defined for unweighted complete graphs even if the chromatic correlation clustering problem is defined in general for arbitrary graphs. The authors do not bring up this issue, nor explain how (if?) their approach could be generalized to arbitrary graphs (would the same approximation ratio hold in this case?). Note that here there is no generic cost objective function so that it would suffice to add edges of zero costs, and in this way trivially transform any graph in a complete graph.

Below are further comments:
1. The first sentence in the abstract sounds a bit as if Bonchi et. al. had introduced regular CC.
2. In the last sentence of the abstract, "efficiency of our proposed algorithms", I would remove the plural since experiments are included for one of the two algorithms only.
3. In the first part of the introduction where the problem is introduced informally, a reader could get the impression that a multigraph is considered whenever there exist two types of relationships between two given nodes.
4. Page 2, line 49: The authors mention several times that CC has an integrality gap of 2, and that therefore this transfers to the chromatic correlation clustering. However, they are actually only talking about approximations obtained by rounding some LP relaxation. If one were not to use LP relaxation in order to derive approximations algorithm, this is not necessarily true anymore. In fact, having an integrality gap of 2 does not imply that there cannot be an approximation algorithm with a better guarantee than 2 in general. The authors should be more open and clear about it.
5. Page 2, line 72: it would be nice if the meaning of gamma was described with words. From my understanding, an edge labeled with gamma is a "must cut" edge. I would suggest the authors to provide this intuition sometime before equation (1).
6. Page 2 (and everywhere it is stated): CCC-LP cannot be a formulation of the chromatic correlation clustering. In fact, an optimal solution to CCC-LP is in general fractional and cannot correspond to a clustering. An INTEGER solution does, not a fractional. Therefore, if the authors were to add binary constraints to CCC-LP, then they would have a proper formulation of the chromatic correlation clustering. Currently, the LP model they propose, CCC-LP, is just a relaxation of the chromatic correlation clustering, not a formulation.
7. Page 2, starting from line 82 and then a bit everywhere in the paper: the authors should add the corresponding quantifiers whenever they write x^c_{uv}, x^c_u, x^c_v and similar things. Is it for all c,u,v or just a subset? If so, which?
8. Advice about Algorithm 1 line 3, observation 4.1 and related issues: I would make it explicit that for a fixed v there exists a unique c such that x_v^c < 1/2, and I would write this remark early on, as soon as the reader reads the algorithm so that it is clear that line 3 is well-defined.
9. Algorithm 1, line 9: what does "pivot" mean? Is it just any node in S_c or does a node need to satisfy specific properties in order to be a pivot?
10. Page 3, line 101-102: also lines 5,6 in the algorithm provide a partitioning.
11. Page 3, line 108: I would replace "rationality" with something more like "motivation, intuition, idea behind it...".
12. Page 3, lines 109-110: "as to" --> "with respect to"?
13. Page 4, first paragraph: Note that the rounding of x_u^c does not affect the objective value. I do not understand what is supposed to be greedy here.
14. Page 4, line 125: "until every vertex in S_c is included in a cluster" from the way it is written it sounds like all these vertices become singletons, while this is not the case since you grow these clusters C_k.
15. Page 4, lines 130-133: the authors should mention where the proof can be found. Note that the complete proof is not available in this manuscript since part of it has been moved to the appendix.
16. Page 4, line 145: what is LP? CCC-LP?
17. Page 5, line 158: write "some alpha >= 2" to emphasize that in inequality does not need to hold for every alpha. Observe that if it were for all alphas, then it would hold also for alpha = 2 and we would have a 2-approximation.
18. Page 5, line 175: it looks to me like E^t_c should be E^t_c \setminus E^{t+1}_c.
19. End of page 5: Why is it necessary to argue with submartingales? Here, we are dealing with an expected value of a finite sum and the linearity of the expected value should be all that is necessary for this argument.
20. Algorithm 2: Why is not Output described here? See Algorithm 1.
21. Algorithm 2, line 3: "a set" not "the set".
22. Algorithm 2: see second question in section "3. Questions".
23. Algorithm 2, line 7: One can get the impression that being a "seed" vertex is a special property. But by my understanding every vertex in "S_k" is a seed vertex. For me, it would be easier to understand if it just said "Let each vertex in S_k vote...".
24. Algorithm 2: Why are the clusters called S instead of C as above?
25. Page 7, line 220: I would introduce this notation before so that the reader is not confused when he/she first reads it in the fourth line of the algorithm.
26. Page 8, line 241: It would be nice to see how the constant m affects the runtime.
27. Checklist 1b: I do not see a discussion of the limitations neither in the abstract nor in section 6.
28. Checklist 3c: the error bars are not present in the main paper.

---

> ### Author Response · Authors · 2022-08-02
> **Response to reviewer 4VF8 (part 1/3)**
>
> # Reviewer 4VF8:
>
> Thank you very much for your great comments. Please find our detailed responses in the following:
>
> > The approximation algorithm (which should be the main result of this paper) has not been implemented and tested on the data sets. ... In particular, it would have been nice to see how tight the approximation ratio is in practice.
>
> **Response**: Nice suggestion. Unfortunately, we find that solving LP for CCC requires prohibitive amount of memory and running time in practice, which are polynomial in theory though. As a result, we are unable to obtain the results for the LP-based approaches due to equipment limitations. To our knowledge, no existing work provides empirical results for the LP-based approaches including [2, 3, 10, 13, 17, 18], which indicate that such approaches are more of theoretical interests.
>
> > The statement in the abstract about the approximation algorithm is misleading. ... This is in contradiction to the (deterministic) statements in the abstract and introduction.
>
> **Response**: We appreciate your rigorous comment. In the uploaded revision, we have incorporated expectation in the approximation guarantee as advised. Moreover, we have added a new section to show how to derandomize our algorithm in Appendix C of the uploaded revision, so that a worst-case approximation guarantee of 2.5 can be attained.
>
> > The choice of references a bit odd. It would be nice if the vast literature on the clique partitioning/multicut problem was cited. In this literature, the correlation clustering ILP is studied in great detail. See e.g. Groetschel \& Wakabayashi, Chopra \& Rao, Deza \& Groetschel \& Laurent, Bandelt et. al. ... (and references therein).
>
> **Response**: Thanks for mentioning the nice relevant work. In the uploaded revision, we have cited and discussed about these elegant work in Section 1.
>
> > The approximation algorithm (Algorithm 1) is only defined for unweighted complete graphs even if the chromatic correlation clustering problem is defined in general for arbitrary graphs. ... Note that here there is no generic cost objective function so that it would suffice to add edges of zero costs, and in this way trivially transform any graph in a complete graph.
>
> **Response**: According to the definition of CCC by Bonchi et al. [6], the edge with color $\gamma$ actually represents an absent edge. Therefore, CCC transforms any graph in a complete graph by labelling the absent edges with color $\gamma$. In the uploaded revision, we have clarified this to avoid confusion.
>
> > Is the analysis in section 4 tight?
>
> **Response**: It is a nice comment. The same as the LP-based approach in the literature [2, 3, 10, 13, 17, 18], the approximation ratio derived in this paper is based on the gap between our rounded solution and the LP solution, which is looser than that compared to the optimal solution. However, to our knowledge, constructing a tight example for the LP-based solution compared to the optimal solution is rather challenging, which has not been addressed in the existing studies. On the other hand, in what follows, we provide a simple example where the ratio of our rounded solution to the LP solution is 2. Consider a graph with 3 vertices and 3 edges with distinct colors such that $\phi(uv)=1$, $\phi(vw)=2$ and $\phi(wu)=3$. It is easy to get that the LP solution is: $x_u^2=x_v^3=x_w^1=1$, $x_u^1=x_u^3=x_v^1=x_v^2=x_w^2=x_w^3=1/2$, $x_{uv}^{1}=x_{vw}^{2}=x_{wu}^{3}=1/2$ and $x_{uv}^{2}=x_{uv}^{3}=x_{vw}^{1}=x_{vw}^{3}=x_{wu}^{1}=x_{wu}^{2}=1$. Thus, the LP solution has an objective value of $3/2$. Meanwhile, our rounded solution will partition all three vertices as outliers in Phase 1 and classify each vertice as a singleton clustering in Phase 2, where the objective value is $3$, showing a ratio of $2$ compared to the LP solution. In a short, analyzing the tightness of the approximation ratio achieved by the LP-based approach for both CC and CCC is an interesting problem and remains an open question, which will be left as our future work.
>
> > Why are we guaranteed that Algorithm 2 does not get stuck in the first while loop? In fact, as long as $E$ has at least one edge we stay in the loop, however we need at least m edges to perform all the operations inside of the while loop (and then remove edges from $E$). It looks to me like the algorithm could arrive at a point where $E$ is not empty but contains less than $m$ edges. This would result in getting stuck in the loop.
>
> **Response**: Sorry for the confusion caused. Actually we use sampling with replacement to pick the $m$ edges in the first while loop. Therefore, it never gets stuck in the first while loop. In the uploaded revision, We have revised it to avoid confusions.

---

> > ### Author Response · Authors · 2022-08-02
> > **Response to reviewer 4VF8 (part 2/3)**
> >
> > > The first sentence in the abstract sounds a bit as if Bonchi et al. had introduced regular CC.
> >
> > **Response**: Thanks. we have revised it to avoid confusion.
> >
> > > In the last sentence of the abstract, "efficiency of our proposed algorithms", I would remove the plural since experiments are included for one of the two algorithms only.
> >
> > **Response**: Thanks. we have revised it as advised.
> >
> > > In the first part of the introduction where the problem is introduced informally, a reader could get the impression that a multigraph is considered whenever there exist two types of relationships between two given nodes.
> >
> > **Response**:
> > Thanks. we have revised it to avoid confusions.
> >
> > > Page 2, line 49: The authors mention several times that CC has an integrality gap of $2$, and that therefore this transfers to the chromatic correlation clustering. However, they are actually only talking about approximations obtained by rounding some LP relaxation. If one were not to use LP relaxation in order to derive approximations algorithm, this is not necessarily true anymore. In fact, having an integrality gap of 2 does not imply that there cannot be an approximation algorithm with a better guarantee than 2 in general. The authors should be more open and clear about it.
> >
> > **Response**: We fully agree with the reviewer. The integrality gap of LP relaxation we used for CCC is at least 2, which does not indicate that CCC cannot be approximated within a constant factor of 2. However, our algorithm gets a ratio of 2.5 for CCC, which is closed to the best-known 2.06 ratio for CC [10]. Moreover, identifying the inapproximablity of CCC could be hard; even for CC, no meaningful lower bound is known for its achievable approximation ratio. We have revised the uploaded revision accordingly to avoid confusions.
> >
> > > Page 2, line 72: it would be nice if the meaning of gamma was described with words. From my understanding, an edge labeled with gamma is a "must cut" edge. I would suggest the authors to provide this intuition sometime before equation (1).
> >
> > **Response**: Thanks. we have clarified this in the uploaded revision.
> >
> > > Page 2 (and everywhere it is stated): CCC-LP cannot be a formulation of the chromatic correlation clustering. In fact, an optimal solution to CCC-LP is in general fractional and cannot correspond to a clustering. An INTEGER solution does, not a fractional. Therefore, if the authors were to add binary constraints to CCC-LP, then they would have a proper formulation of the chromatic correlation clustering. Currently, the LP model they propose, CCC-LP, is just a relaxation of the chromatic correlation clustering, not a formulation.
> >
> > **Response**: Thanks. We have revised to clarify that CCC-LP is the LP relaxation of the CCC problem.
> >
> > > Page 2, starting from line 82 and then a bit everywhere in the paper: the authors should add the corresponding quantifiers whenever they write $x^c_{uv}, x^c_u, x^c_v$ and similar things. Is it for all $c,u,v$ or just a subset? If so, which?
> >
> > **Response**: Thanks. we have revised it as advised.
> >
> > > Advice about Algorithm 1 line 3, observation 4.1 and related issues: I would make it explicit that for a fixed $v$ there exists a unique $c$ such that $x_v^c < 1/2$, and I would write this remark early on, as soon as the reader reads the algorithm so that it is clear that line 3 is well-defined.
> >
> > **Response**: Thanks for your suggestion. According to our analysis, there may be no vertex $v$ satisfying $x_v^c < 1/2$ for some $c\in L$, resulting in $S_c=\emptyset$, but this does not affect the correctness of our algorithm. On one hand, our algorithm checks if $S_c$ is empty before $S_c$ is clustered at every iteration. On the other hand, all cases of $S_c$ are considered in our analysis. We have revised our paper to clarify this.
> >
> > > Algorithm 1, line 9: what does "pivot" mean? Is it just any node in $S_c$ or does a node need to satisfy specific properties in order to be a pivot?
> >
> > **Response**: Thanks. we have revised it as advised. Actually, any unclustered vertex can be a "pivot".
> >
> > > Page 3, line 101-102: also lines 5,6 in the algorithm provide a partitioning.
> >
> > **Response**: Thanks. Lines 5-6 belong to Phase 2 of Algorithm 1, i.e., clustering the outliers. Also it can be considered as partionioning, as clustering is similar to partitioning.
> >
> > > Page 3, line 108: I would replace "rationality" with something more like "motivation, intuition, idea behind it...".
> >
> > **Response**: Thanks. we have revised it as advised.
> >
> > > Page 3, lines 109-110: "as to" $-->$ "with respect to"?
> >
> > **Response**: Thanks. we have revised these as advised.

---

> > > ### Author Response · Authors · 2022-08-02
> > > **Response to reviewer 4VF8 (part 3/3)**
> > >
> > > > Page 4, first paragraph: Note that the rounding of $x_u^c$ does not affect the objective value. I do not understand what is supposed to be greedy here.
> > >
> > > **Response**: Sorry for the confusion caused. We round the variable $x_u^c$ in order to decide the color of the cluster which includes vertex $u$. And the cluster's color does affect the objective value.
> > >
> > > > Page 4, line 125: "until every vertex in $S_c$ is included in a cluster" from the way it is written it sounds like all these vertices become singletons, while this is not the case since you grow these clusters $C_k$.
> > >
> > > **Response**: Sorry. we have clarified this to avoid confusion in the uploaded revision. Actually, what we want to say is "all vertices have been clustered".
> > >
> > > > Page 4, lines 130-133: the authors should mention where the proof can be found. Note that the complete proof is not available in this manuscript since part of it has been moved to the appendix.
> > >
> > > **Response**: Thanks. we have revised these as advised.
> > >
> > > > Page 4, line 145: what is LP? CCC-LP?
> > >
> > > **Response**: Sorry. we have clarified this to avoid confusion in the uploaded revision. It should be LP cost (the objective function value of CCC-LP).
> > >
> > > > Page 5, line 158: write "some $\alpha >= 2$" to emphasize that in inequality does not need to hold for every alpha. Observe that if it were for all alphas, then it would hold also for $\alpha = 2$ and we would have a 2-approximation.
> > >
> > > **Response**: Thanks for your reminder and we have revised it.
> > >
> > > > Page 5, line 175: it looks to me like $E^t_c$ should be $E^t_c \setminus E^{t+1}_c$.
> > >
> > > **Response**: Thanks. Since we only consider the events that $\epsilon_u$ and $v$, an edge $uv\in E^t_c$ will affect $D_t$ only if $uv\in E^t_c \setminus E^{t+1}_c$. Therefore, $E^t_c$ or $E^t_c \setminus E^{t+1}_c$ are both ok.
> > >
> > > > End of page 5: Why is it necessary to argue with submartingales? Here, we are dealing with an expected value of a finite sum and the linearity of the expected value should be all that is necessary for this argument.
> > >
> > > **Response**: Thanks. As the stopping time T is randomized, we have used submartingales for the argument.  But your suggestion is very interesting and we will study it.
> > >
> > > > Algorithm 2: Why is not Output described here? See Algorithm 1.
> > >
> > > **Response**: Sorry for missing it and appreciate your reminder. We have added the Output in Algorithm 2.
> > >
> > > > Algorithm 2, line 3: "a set" not "the set".
> > >
> > > **Response**: Thanks for your reminder. We have revised it.
> > >
> > > > Algorithm 2: see second question in section "3. Questions".
> > >
> > > **Response**: Thanks. Please see our response to your questions.
> > >
> > > > Algorithm 2, line 7: One can get the impression that being a "seed" vertex is a special property. But by my understanding every vertex in "$S_k$" is a seed vertex. For me, it would be easier to understand if it just said "Let each vertex in $S_k$ vote...".
> > >
> > > **Response**: Thanks. We have revised it as advised.
> > >
> > > > Algorithm 2: Why are the clusters called $S$ instead of $C$ as above?
> > >
> > > **Response**: Sorry for the confusion caused. There are actually two sets $S$ (seed set) and $C$ (candidates set) in our analysis for Algorithm 2. So we differentiate $S$ from $C$.
> > >
> > > > Page 7, line 220: I would introduce this notation before so that the reader is not confused when he/she first reads it in the fourth line of the algorithm.
> > >
> > > **Response**: Thanks for your suggestion. We have revised it.
> > >
> > > > Page 8, line 241: It would be nice to see how the constant $m$ affects the runtime.
> > >
> > > **Response**: Thanks. We have followed your suggestion to study the impact of m and add new experimental results. Details can be found in Appendix D of the uploaded revision.

---

### Official Review · Reviewer_7M1G · 2022-07-07

**Rating:** 5
**Confidence:** 3
**Soundness:** 3 good
**Presentation:** 3 good
**Contribution:** 3 good

**Summary:**

The authors develop a 2.5-approximation algorithm for chromatic correlation clustering (CCC), which is a generalization of the well-known correlation clustering problem introduced in 2012. Their algorithm is LP-based, while it improves the best-known approximation ratio of 3 developed in 2021. A greedy heuristic is also developed and compared against state-of-the-art algorithms on real-world data, showing promising results.

**Questions:**

see my comments above.

**Limitations:**

There doesn't seem to be any negative societal impact.

**Strengths And Weaknesses:**

+ the authors develop the best approximation algorithm for chromatic correlation clustering. In contrast with the previous best 3-approximation algorithm, their algorithm is not color blind, which is interesting.
+ the experimental evaluation on real-world data is promising (although still preliminary)
+ the paper is well written and organized

- the theoretical results are not surprising with some of the ideas already presented in [2] and [3]
- it is not clear whether the analysis of their algorithm is tight. That is, is there an example where their algorithm produces solutions with an approximation ratio of 2.5?
- the experimental evaluation is not convincing. As most of the algorithms are randomized, the authors should report the variance of their results. As their results are very close to the ones of [20], there might not be much difference, statistically. Also it seems that the parameters of their algorithms have been optimized w.r.t. to the datasets considered (eps and m have been set to 0.55 and 10 respectively), while the default settings for the other algorithms have been used. It is not clear why, the Cooking dataset from [20] has not been included in their experimental evaluation. This does not seem to be a fair evaluation.
- the code of their algorithm does not seem to be publicly available

some other comments:
- “It is noted that CCC is a generalized version of CC and hence also has an integrality gap of 2 as the lower bound for  any achievable approximation ratio for it. “ This claim is ambiguous. It should be noted that this lower bound holds only for the LP-formulation considered in the paper. This claim alone does not prevent the existence of another LP-formulation with smaller integrality gap or a completely different algorithm with approximation guarantee < 2.
- there seems to be some confusion between randomly and arbitrarily (e.g. line 116)
- theorem 3.1, it should be stated that the approx is in expectation.
- which LP-solver has been used in the experimental evaluation?

Overall the paper contains some interesting ideas, while it has strong potential. In particular they provide the best approximation algorithm for CCC. However, I feel that the points discussed above should be addressed before being accepted for publication.

---

> ### Author Response · Authors · 2022-08-02
> **Response to reviewer 7M1G (part 1/2)**
>
> # Reviewer 7M1G:
>
> Thank you very much for your great comments. Please find our detailed responses in the following:
>
> > the theoretical results are not surprising with some of the ideas already presented in [2] and [3].
>
> **Response**: We would like to clarify that LP-based approach is commonly used by existing work for CC [2, 10, 13, 17, 18, 20, 21] and CCC [3], and the key challenges and contributions lie in how to round the LP solution such that better approximation guarantees can be achieved. Specifically, Ailon et al. [2] proposed an LP-rounding based algorithm with an approximation ratio of 2.5 for CC (which a special case of CCC when $|L|=1$), and Chawla et al. [10] improved this ratio to 2.06 by contributing a more sophisticated LP-rounding technique. Different from them, we study a generalized problem of CCC, where deriving a rounding approach with theoretical guarantees is much more challenging as demonstrated in previous work [3].
>
> In addition, Anava et al. [3] proposed an LP rounding approach for CCC with an approximation ratio of 4, and we develop a simple yet effective LP rounding algorithm that achieves an improved ratio of 2.5. In particular, we devise a novel two-phase rounding procedure, which is very different from that by Anava et al. [3]. That is, we first classify all vertices of the graph into several groups on the basis of $x_v^c$ (with the vertices in each group colored the same), and then cluster the nodes in each group according to $x_{uv}^c$. In contrast, Anava et al. [3] repeatedly choose a vertice $v$ at random as the center of a new cluster and include a set of vertices that are close to $v$ if certain condition is met, which only relies on $x_{uv}^c$. To summarize, we propose a novel rounding approach for CCC with an approximation ratio of 2.5 through a non-trivial theoretical analysis, which improves the best-known factor of 4 by Anava et al. [3] (LP-based) and 3 by Klodt et al. [24] (non-LP-based). Therefore, we believe that our theoretical contributions are significant due to the novel LP rounding algorithm and the non-trivial analysis.
>
> > it is not clear whether the analysis of their algorithm is tight. That is, is there an example where their algorithm produces solutions with an approximation ratio of 2.5?
>
> **Response**: It is a nice comment. The same as the LP-based approach in the literature [2, 3, 10, 13, 17, 18], the approximation ratio derived in this paper is based on the gap between our rounded solution and the LP solution, which is looser than that compared to the optimal solution. However, to our knowledge, constructing a tight example for the LP-based solution compared to the optimal solution is rather challenging, which has not been addressed in the existing studies. On the other hand, in what follows, we provide a simple example where the ratio of our rounded solution to the LP solution is 2. Consider a graph with 3 vertices and 3 edges with distinct colors such that $\phi(uv)=1$, $\phi(vw)=2$ and $\phi(wu)=3$. It is easy to get that the LP solution is: $x_u^2=x_v^3=x_w^1=1$, $x_u^1=x_u^3=x_v^1=x_v^2=x_w^2=x_w^3=1/2$, $x_{uv}^{1}=x_{vw}^{2}=x_{wu}^{3}=1/2$ and $x_{uv}^{2}=x_{uv}^{3}=x_{vw}^{1}=x_{vw}^{3}=x_{wu}^{1}=x_{wu}^{2}=1$. Thus, the LP solution has an objective value of $3/2$. Meanwhile, our rounded solution will partition all three vertices as outliers in Phase 1 and classify each vertice as a singleton clustering in Phase 2, where the objective value is $3$, showing a ratio of $2$ compared to the LP solution. In a short, analyzing the tightness of the approximation ratio achieved by the LP-based approach for both CC and CCC is an interesting problem and remains an open question, which will be left as our future work.
>
> > the experimental evaluation is not convincing. As most of the algorithms are randomized, the authors should report the variance of their results. As their results are very close to the ones of [20], there might not be much difference, statistically.
>
> **Response**: Following your suggestion, we have reported the variances of the implemented algorithms. Details can be found in Appendix D of the uploaded revision. In summary, the experimental results show that the variances of our GV algorithm are smaller than those of the competitors on most datasets. In particular, compared to the GE algorithm which has the second best performance on the objective function value, the variances of GV are smaller on 9 out of 10 datasets, which demonstrates the robustness of our method.

---

> > ### Author Response · Authors · 2022-08-02
> > **Response to reviewer 7M1G (part 2/2)**
> >
> > > Also it seems that the parameters of their algorithms have been optimized w.r.t. to the datasets considered (eps and m have been set to 0.55 and 10 respectively), while the default settings for the other algorithms have been used.
> >
> > **Response**: To tradeoff the efficiency and effectiveness, we empirically set $\epsilon=0.55$ and $m=10$ in the experiments. Meanwhile, for the baselines, we follow all the parameter settings (if any) according to their proposals (which are likely to be optimized by the proposers). To evaluate the impacts of the parameters, we report the results when the settings of $m$ are changed. The results show that increasing the parameter $m$ increases the running time, while NOD will decrease with $m$ increasing. More details about this can be found in Appendix D of the uploaded revision.
> >
> > > It is not clear why, the Cooking dataset from [20] has not been included in their experimental evaluation. This does not seem to be a fair evaluation.
> >
> > **Response**: The result on the Cooking dataset is consistent with those observed in the reported results. Sorry for missing this result and appreciate your reminder. We have added the experimental results on the Cooking dataset, which show that our GV algorithm still outperform the other baselines in terms of the objective function value on the Cooking dataset. Please refer to Section 6 of our uploaded revision for details.
> >
> > > the code of their algorithm does not seem to be publicly available
> >
> > **Response**: Thanks. We have included all code in the supplementary materials of this submission. As this paper is not published yet, we will make the code publicly available once the paper is published.
> >
> > > "It is noted that CCC is a generalized version of CC and hence also has an integrality gap of $2$ as the lower bound for any achievable approximation ratio for it. " This claim is ambiguous. It should be noted that this lower bound holds only for the LP-formulation considered in the paper. This claim alone does not prevent the existence of another LP-formulation with smaller integrality gap or a completely different algorithm with approximation guarantee $< 2$.
> >
> > **Response**: We fully agree with the reviewer. The integrality gap of LP relaxation we used for CCC is at least 2, which does not indicate that CCC cannot be approximated within a constant factor of 2. However, our algorithm gets a ratio of 2.5 for CCC, which is closed to the best-known 2.06 ratio for CC [10]. Moreover, identifying the inapproximablity of CCC could be hard; even for CC, no meaningful lower bound is known for its achievable approximation ratio. We have revised the uploaded revision accordingly to avoid confusions.
> >
> > > there seems to be some confusion between randomly and arbitrarily (e.g. line 116).
> >
> > **Response**:Thanks for pointing out it. We have revised it to "arbitrarily" to keep consistence.
> >
> > > theorem 3.1, it should be stated that the approx is in expectation.
> >
> > **Response**: We appreciate your rigorous comment, and have incorporated expectation in the approximation guarantee as advised. Moreover, we add a new section to show how to derandomize our algorithm in Appendix C of the uploaded revision, so that a worst-case approximation guarantee can be attained.

---

### Official Review · Reviewer_77Xk · 2022-07-10

**Rating:** 6
**Confidence:** 3
**Soundness:** 3 good
**Presentation:** 3 good
**Contribution:** 2 fair

**Summary:**

This paper introduces a new 2.5-approximation LP-based algorithm for Chromatic Correlation Clustering improving upon existing state-of-the-art 3.0-approximation result of Klodt et al. The authors also present a two-phase greedy algorithm (without guarantees) and demonstrate empirically that it performs better in terms of clustering found (minimizing number of disagreements) with improved time complexity compared to previous state-of-the-art solution of Klotz et al.


**Questions:**

N/A

**Strengths And Weaknesses:**

## Strengths
+ Well-written with clear presentation of main results.
+ New rounding procedure based on the LP solution with careful analysis leading to 2.5-approximation compared to previous state-of-the-art theoretical result of Klotz et al.
+ GreedyVote algorithm which performs better empirically compared to previous methods on standard CCC datasets.

## Weaknesses
- Limited impact for wider Neurips audience - More suited to KDD/WWW possibly?

---

> ### Author Response · Authors · 2022-08-02
> **Response to reviewer 77Xk (part 1/1)**
>
> # Reviewer 77Xk:
>
> Thank you very much for your great comments. Please find our detailed responses in the following:
>
> > Limited impact for wider Neurips audience - More suited to KDD/WWW possibly?
>
> **Response**: We agree on that Correlation Clustering (CC) and Chromatic Correlation Clustering (CCC) are well suited for KDD/WWW, but in our humble opinion, they are also a good fit for NeurIPS as they are extensively studied in the community. In particular, from 2018 to 2021, NeurIPS and ICML have published at least 10 research papers [7, 8, 20, 21, 23, 25, 26, 30, 31, 32] on developing (approximation) algorithms for CC/CCC (including the variants).

---

### Meta-Review · Area_Chair_nBVx · 2022-08-24

**Recommendation:** Accept
**Confidence:** Certain

**Metareview:**

The papers studied that Chromatic Correlation clustering problem, the author(s) introduce(s) a new 2.5-approximation LP-based algorithm  improving upon existing state-of-the-art.

The result is not particularly novel but the paper contains a new algorithm for a well-studied problem and prove that ideas inspired from their theory are impactful in practice so it would be a nice contribution to the NeurIPS program.

The committee suggestion is to accept the. paper as a poster.

**Award:**

No

---

### Decision · Program_Chairs · 2022-09-14

Accept